# Situation Analysis of Early Implementation of Programmatic Management of Tuberculosis Preventive Treatment among Household Contacts of Pulmonary TB Patients in Delhi, India

**DOI:** 10.3390/tropicalmed9010024

**Published:** 2024-01-17

**Authors:** Yasir Alvi, Sairu Philip, Tanu Anand, Palanivel Chinnakali, Farzana Islam, Neeta Singla, Pruthu Thekkur, Ashwani Khanna, BK Vashishat

**Affiliations:** 1Department of Community Medicine, Hamdard Institute of Medical Sciences and Research, New Delhi 110062, India; drfarzanaislam@gmail.com; 2Department of Community Medicine, Government Medical College, Kottayam 686008, India; sairuphilip2018@gmail.com; 3Scientist E, Indian Council of Medical Research, Department of Health Research, Ministry of Health and Family Welfare, New Delhi 110029, India; tanu.anand@icmr.gov.in; 4Department of Preventive and Social Medicine, Jawaharlal Institute of Postgraduate Medical Education and Research (JIPMER), Puducherry 605006, India; palaniccm@gmail.com; 5Department of Training, National Institute of TB & Respiratory Disease, New Delhi 110030, India; docneetasingla@gmail.com; 6Centre for Operational Research, International Union Against Tuberculosis and Lung Disease (The Union), 75006 Paris, France; pruthu.tk@theunion.org; 7Delhi State NTEP, New Delhi 110002, India; drashwani.khanna@gmail.com; 8State TB Cell, Gulabi Bagh, New Delhi 110007, India; stodl@rntcp.org

**Keywords:** latent tuberculosis infection, isoniazid preventive therapy (IPT), contact tracing, care cascade, SORT-IT, Operational Research (OR)

## Abstract

Tuberculosis Preventive Treatment (TPT) is a powerful tool for preventing the TB infection from developing into active TB disease, and has recently been expanded to all household contacts of TB cases in India. This study employs a mixed-methods approach to conduct a situational analysis of the initial phase of TPT implementation among household contacts of pulmonary TB patients in three districts of Delhi, India. It was completed using a checklist based assessments, care cascade data, and qualitative analysis. Our observations indicated that organizational structure and planning were established, but implementation of TPT was suboptimal with issues in drug availability and procurement, budget, human resources, and training. Awareness and motivation, and shorter regimen, telephonic assessment, and collaboration with NGOs emerged as enablers. Apprehension about taking TPT, erratic drug supply, long duration of treatment, side effects, overburden, large population, INH resistance, data entry issues, and private provider reluctance emerged as barriers. The study revealed potential solutions for optimizing TPT implementation. It is evident that, while progress has been made in TPT implementation, there is room for improvement and refinement across various domains.

## 1. Introduction

India is a high-tuberculosis (TB)-burden country, both in terms of the number of people with the disease and infection. It is estimated that 350 to 400 million Indians are infected with Mycobacterium tuberculosis, which is approximately one-third of the total population of the country [1,2]. However, only a small fraction with the TB infection (TBI) progress to full-fledged TB disease; this is estimated at 2.6 million patients annually [1,3,4]. Among these individuals with TBI, approximately 5% progress into active disease within 2 years [5]. The risk of developing active TB disease varies with several factors, with high susceptibility among people living with HIV (PLHIV) and individuals with weakened immune systems [4,6].

The primary source of TBI is close contact with an individual suffering from pulmonary TB disease, typically a household contact. To reduce the chances of developing TB disease among household contacts with TBI, the World Health Organization (WHO) recommends providing TB preventive treatment (TPT) after ruling out active disease [7]. TPT is also a key component of Pillar 1 of the WHO End TB Strategy, with a global target to achieve 90% TPT coverage among PLHIV and household contacts of pulmonary TB patients’ by 2025 [7]. TPT effectively reduces the risk of disease and mortality among various priority groups [8,9,10,11]. Despite the global commitment and efforts, TPT access and uptake among persons at risk remains low [12].

In India, TPT in the form of Isoniazid preventive therapy (IPT) for child household contacts of pulmonary TB patients was implemented two decades ago. With making substantial progress in tuberculosis diagnosis and treatment, there arises a need for focus on preventive strategies [2]. Subsequently, with the firm commitment of Government of India (GoI) to end TB by 2025, following the WHO recommendations, the TPT was expanded to all the household contacts in 2021 under the renewed National Tuberculosis Elimination Program (NTEP) [13,14]. The state of Delhi followed suit and implemented TPT under mission mode utilizing the public–private partnership (PPP) in some districts, while operating as a routine service in others. Escalating the initiative, the GoI in 2021 adopted a ‘cascade of care’ approach as a core strategy for delivering TPT services to all priority groups across India, including household contacts of pulmonary TB patients [13].

The cascade of care provides a valuable framework for assessing and improving TPT implementation. It tracks the sequential steps (cascade), capturing information on contact tracing, screening for symptoms, evaluation for TB disease, initiation and completion of TPT [13]. Contact tracing often undertaken with screening, involves a healthcare worker visiting the household of a TB index patient to identify and list all household contacts (HHCs) and assess them for symptoms suggestive of TB. This has been integral to TB programs for decades and is now a core indicator for monitoring progress towards the End TB Strategy [7]. Following ruling out active TB and TPT complications, eligible HHCs are initiated on TPT. Regular counseling, follow-up, and monitoring of treatment adherence and TPT outcomes are crucial components of this cascade. By analyzing each step in the cascade, we can not only identify performance gaps but also implement timely intervention to optimize TPT effectiveness and its impact.

Globally, TPT implementation has been a challenge in programmatic settings [12]. Despite limited data on TPT among all household contacts, existing studies among children do provide some estimates. It has been observed that the performance of initial care cascade indicators are satisfactory; TPT initiation and completion rates among HHC are subpar [15]. Studies in India revealed suboptimal implementation of the erstwhile IPT program, prompting various interventions to improve its uptake and effectiveness [16,17,18,19]. A similar trend is observed in other countries as well [20,21,22]. Ideally, the implementation of TPT for all household contacts since January 2022 should have been smooth, considering the prior experiences of implementing IPT. However, there is limited information from India on the programmatic implementation of TPT for all household contacts. In this study, we conduct a situation analysis of the initial phase of programmatic management of TPT among household contacts of pulmonary TB patients registered between 1 January to 30 June 2022 at a selected district TB center (DTC) in Delhi, with an aim to assess the implementation and observe the cascade of TPT care, and explore the enablers, barriers, and possible solutions.

## 2. Materials and Methods

### 2.1. Study Design

It was a mixed-method study (convergent design), with quantitative (routinely collected secondary programmatic data) and qualitative (descriptive study design) components.

### 2.2. Setting

#### 2.2.1. General Setting

The study was conducted in Delhi, the capital of India, with a population of about 16.8 million [23]. (Appendix A) Due to ample job opportunities, its population is predominantly urban and migrants. Administratively, it is a Union territory comprising 11 districts. Delhi is India’s highest TB-burden state for all forms of TB, with 747 patients per 100,000 of the population [4].

#### 2.2.2. Specific Setting

The District Tuberculosis Center (DTC), locally known as the chest clinic, is the administrative unit for TB care in Delhi. There are 25 DTC, subdivided into Tuberculosis Units (TU) for every 5 lakh population. Every TU is further divided into Designated microscopy centers (DMC) and Peripheral Health Institutions, commonly known as DOTS centers, manned by a DOT provider.

TPT is being implemented in all the 25 DTCs. Six DTCs are implementing TPT under public–private partnership (PPP) project and dedicated TPT staff supported by the project carry out contact tracing and regular monitoring, while drugs are dispensed by existing staff. In 19 DTCs, the implementation is under routine program setting with existing staff carrying out all the activities. A screening and ‘treat only’ strategy is being used to provide TPT among beneficiaries in Delhi in 19 DTC, while a ‘test and treat’ strategy, utilizing a Mantoux test, is being utilized at 6 DTC. Universally, isoniazid for six month (6H) is being provided to the TPT beneficiary; although a shorter treatment regimen, a combination of isoniazid and rifapentine for three month (3HP) was also available at a few centers.

Once a pulmonary TB patient is identified/diagnosed, utilizing contact tracing, all the household contacts are screened for active disease with clinical, laboratory, and radiological assessment, depending on which strategy is being followed. (Appendix A) At the DTC under PPP, initial 4S symptoms screening (cough, fever, night sweats, and weight loss), treatment initiation support, and regular counselling were carried out by the dedicated TPT project staff while the linked DOTS center dispenses drugs. The DTC where TPT was implemented under routine TB care, screening, treatment, drug dispensation, and regular counselling was completed utilizing staff of the DOTS center. The DOT provider and field coordinators were responsible for collecting the data, which are monitored by the program manager at each district and subsequently uploaded over the TB information system of India known as Nikshay. Along with this, in the DTCs under PPP, the project staff were separately collected the data and uploaded them at the project portal.

This study was to be conducted at three selected DTC; one under PPP, while the other two DTC were under routine program settings. Among the remaining two, one was utilizing a “test and treat’ strategy using a Mantoux skin test along with an X-ray for diagnosing TB infection before initiating TPT.

### 2.3. Study Population

#### 2.3.1. For the Quantitative Phase

All household contacts (HHC) of newly registered pulmonary TB patients (index patients) at the three selected DTC of Delhi between 1 January to 30 June 2022 in Delhi were included. All HHC who were living with a pulmonary TB patient, registered during the study period, were included after ruling out active TB disease.

#### 2.3.2. For the Qualitative Phase

We interviewed purposively selected 19 healthcare providers (state level TB officer—1, district TB officers—3, medical officers—3, subdistrict program managers—5, community-level treatment providers—5, project staff—2), until data saturation. We did not have any refusal to participate, and no repeat interviews were carried out.

### 2.4. Data Variables and Data Collection

#### 2.4.1. Quantitative Component

The data on socio-demographic characteristics of the HHC (age and gender), and the clinical characteristics (index patients treatment category and date of diagnosis, HHC’s symptoms, date of start of TPT, date of completion or stoppage of TPT) were obtained from Nikshay. (See Appendix A)

For assessing the status of TPT implementation, we adapted WHO’s Service Availability and Readiness Assessment and merged it with Guidelines for Programmatic Management of TB Preventive Treatment in India and prepared a semi-structured TPT checklist [13,24]. It had six domains—Organization Structure, Human Resource and Training, Community Engagement, Diagnostics, Drugs and Logistics, and Private Provider Engagement. (See Table 1) It was reviewed by authors and program managers and was found to be appropriate for assessment. Using the checklist, information from the state TB officer and all district TB officers were collected.

#### 2.4.2. Qualitative Component

Healthcare provider perception regarding implementation of TPT was explored through a key informant interview. The principal investigator (YA), a male medical doctor fluent in the local language (Hindi) and field investigator, male, both trained in qualitative research methods conducted the interviews. Face-to-face interviews after prior intimation and as per the convenience of the participant were planned at their workplace, in seclusion. A semi-structured interview guide was used with broad open-ended questions, and permission to audio record was sought. At the same time, a field investigator captured the conversation, tone, and nature of the dialogue through field notes. In those interviews where permission to record was not granted, data capture was based on field notes. After the interview was over, which lasted 20–35 min, the interviewee was provided the summary of the interview to ensure validation.

For details on data variables, sources of data and data collection, please see Appendix A and for operational definition please see Appendix A.

### 2.5. Data Entry and Analysis

Quantitative data: The source of the data is the routine programmatic data of TPT captured retrospectively from Nikshay. After taking the required permission from the state TB office, the selected data variables from the TPT data were downloaded in Microsoft Excel, de-identified, and cleaned. The TPT checklist was used to collect information on the current status of TPT from the selected district. The categorical variables among sociodemographic, clinical, treatment characteristics, and TB-care cascade variables (Home visit, 4S screening, HHC with TB Symptom, Evaluated for TB, Diagnosed TB, Initiated on TB treatment, and Eligible and Provided TPT) was represented in percentage (%).

Qualitative data: After the interview, it was transcribed and translated on the same day using an audio recording or verbatim field notes. Two investigators (SP and TA), both female medical doctor, analyzed separately the qualitative data using manual descriptive content analysis and thematic analysis under the broad topics: enablers, challenges, and suggested solutions. First, they were repeatedly read, and then they coded portions of the transcripts together for complete immersion. Then, they were group coded into sub-themes, and organized under themes. The third investigator (PC) reviewed the analysis and the discrepancies were discussed and resolved by consensus. After the final qualitative data analysis, the tables were shared with the stakeholders for their feedback and approval. We used a Consolidated Criteria for Reporting Qualitative Research checklist for reporting [25].

Reflexivity statement: All the authors acknowledge that their perspectives, rooted in medical expertise and experience in TB care, may influence data collection, analysis, and interpretation. At the outset of this study, while the authors shared a belief in the potential of TPT to improve TB outcomes, they also recognized the existence of significant barriers in successful implementation. They recognize their prior experiences as medical doctors, including the direct involvement of a few authors in TB control, could introduce biases. To mitigate this, they employed refutational analysis techniques, engaged in regular self-reflection, and actively sought diverse perspectives throughout the research process. They are committed to maintaining objectivity and transparency, aiming to deliver a rigorous and balanced analysis of TPT implementation challenges and opportunities in Delhi’s context.

### 2.6. Ethics

The study was approved by the Institutional ethics committee, Hamdard Institute of Medical Sciences and Research, New Delhi, India; the Ethics Advisory Group of the International Union Against Tuberculosis and Lung Disease, Paris, France, before conducting the study; and state Operational Research committee, Delhi. Informed consent was recorded prior to the interview and after explaining the objectives of the study in the local language (Hindi). Appropriate administrative permission from the state was sought for extracting the routine data, which was kept in a password-protected computer after removing any patient’s identifiers. Confidentiality of identity and information provided by the study subjects was assured. The participants were provided with feedback on the findings after the study.

## 3. Results

Of the three district TB centers (DTCs), one was implementing TPT services using PPP in project mode, while other two DTC were utilizing a governmental pre-exiting health system. The cascade of TPT care at these centers is shown in Table 1. We observed that despite higher patient load in the project mode district it performed better in contact tracing with initial assessment, enlisting almost all of household contacts (HHC) (97.4% vs. 47.7% and 68.6%), those with symptoms (5.4% vs. 2.0% and 2.8%) and, subsequently, allowing a majority of them to be tested for TB disease (48.0% vs. 28.1% and 35.3%). On contrast, the government-run center performed better in the initiation of TB treatment (100% and 80.0% vs. 64.0%), as well as TPT (68.6% and 97.8% vs. 16.8%). In the ‘Test and Treat’ center, the overall performance was better than in the treat-only government-run center. Additionally, the completion rate of TPT could not be assessed as TPT registers were not updated and linked with contact tracing registers. Apart from the NIKSHAY data compiled by government staff, project staff compiled the TPT project data separately, as show in Table 1 for comparison.

The checklist-based assessment of TPT implementation status in the three DTC is shown in Table 2. We observed that the organizational structure and planning for TPT have been established in the state, but there is a need for budget-allocation refinement, enhanced advocacy for better coverage, and private providers engagement expansion. While the human resource utilization and training are partly complete, there is a requirement for frequent and specialized training, and improved data management in the NIKSHAY portal.

### 3.1. Enablers

A total of 19 interviews were conducted with health providers focused on enablers and challenges in implementation of TPT. One general theme that emerged from the thematic analysis was adequate awareness of TB and TPT among the patients and their household members, which helped them to implement TPT services. One DOT provider remarked, “*ever since we have started this program, the public awareness is very high….people have come forward to take account of it*”. After mixing and analyzing qualitative and quantitative data it was quite apparent that having a dedicated staff for TPT was beneficial for contact tracing and screening. The other key enablers were motivation to take TPT, regimen with better compliance, regular counselling, telephonic assessment, and collaboration with NGOs. Senior Treatment Supervisors (STS) said, “*Counseling is very important; they have to be motivated in such a way that they continue (treatment) completely. We meet every month (for counseling)*”. Another participant expressed, *“… an NGO, it also had a call center. So, calls from the call center goes to every patient’s home. It increased awareness*”. (Figure 1).

### 3.2. Barriers

Major barriers came out from the mixed-method analysis and included erratic drug supply, high patient load, apprehension about taking TPT when not sick, long duration of treatment, side effects, data entry issues, overburden, drug procurement challenges, isoniazid resistance, and private provider reluctance. Barriers can be observed from the quotes in Table 3.

### 3.3. Possible Solutions

During analysis, several potential areas for improvement emerged. Firstly, effective counselling and education were identified as critical components in promoting Tuberculosis Prevention Treatment (TPT) adherence. Participants emphasized the need for comprehensive education campaigns targeting both patients and community members. Additionally, generating accurate and reliable data could help in dispelling misconceptions and enhance understanding of TPT’s preventive benefits. Participants highlighted the need to recruit new staff to meet the increased workload, particularly with the introduction of TPT. The shorter 3HP regimen was proposed as a potential solution to TPT acceptance and adherence. Moreover, it was emphasized that efforts should be directed towards eliminating stigma associated with TB. Private provider engagement emerged as a crucial area for collaboration, suggesting the need for strategic partnerships to broaden the reach of TB prevention efforts. Table 3 enlists possible solutions, and recommendations are mentioned in Table 4.

## 4. Discussion

We conducted a situation analysis of the initial phase of programmatic management of TPT among household contacts of pulmonary TB patients in Delhi. Our findings indicate that organizational structure and planning for TPT were established; however, implementation of TPT was suboptimal, with issues in drug availability and procurement, budget, human resource, and training. Awareness and motivation for TPT uptake among index patients and their household contacts, availability of shorter regimen with better compliance, telephonic assessment, and collaboration with NGOs emerged as enablers. Apprehension about taking TPT, erratic drug supply and procurement challenges, long duration of treatment, side effects, overburden, large population, INH resistance, data entry issues, and private provider reluctance present as barriers.

### Strength and Limitations

The study had several strengths. Firstly, we used a mixed-methods design to assess TPT implantation, both quantitative and qualitative and provided insights into barriers and possible solutions. Secondly, we adopted WHO’s Service Availability and Readiness Assessment toolkit, and merged it with NTEP PMTPT guidelines to prepare a ready-to-use checklist for assessing TPT implementation. This approach for assessment was not only feasible but it was reproducible in different settings. Thirdly, we adhered to NTEP standardized operational definitions for universality. Lastly, we had a comparison group, both within the district (government and project data) and across different implementation strategies for better interpreting the findings.

Nonetheless, there were some limitations. The study was conducted only in three district TB centers in Delhi, selected by the senior program manager, potentially affecting external validity. Secondly, the qualitative data was transcribed from verbatim notes and translated from Hindi to English, which could impact validity and transferability. Thirdly, quantitative analysis was reliant on data from the Nikshay portal, which displayed inaccuracies and incomplete entries due to a lack of training during the program’s early phase. TPT-adherence and -completion rates could not be assessed for the same reason. Further, with subsequent training, our findings might not corresponds to the current scenario. Lastly, the project was time-bound and was limited to healthcare worker’s perspectives; patients’ and their family members’ experiences and expectations could not be assessed, which could inform program improvements.

For assessment of TPT implementation in our study, we developed a checklist to evaluate organization, resources, training, community engagement, diagnostics, drugs, logistics, and private providers engagement [13,24]. It revealed a well-established organizational structure, reflecting recent government’s focus on TPT and prior IPT experience. However, room for improvement exists in terms of coordination and prioritization. The existing state TPT committee needs to communicate priorities to mid-level managers and emphasize TPT in TB-program-review meetings. While the TPT plan is part of the annual Program Implementation Plan (PIP), a separate budget allocation is recommended to ensure dedicated resources for its successful utilization and execution. We also observed private providers’ engagement to be fruitful and recommend optimizing public–private mix (PPM) strategies, especially in underperforming or high-load centers, to improve TPT implementation [26]. Since, TPT was recently implemented in India, we believe this checklist would be a useful tool for assessment in other settings.

The cascade analysis proved invaluable for assessing TPT-care processes. Notably, the district under PPP, despite a higher patient load, outperformed others in contact tracing. Possible reasons could be the dedicated NGO project staff, access to chest X-rays and heightened awareness. One medical officer stated “working with NGOs” and “awareness about TPT” for successful TPT implementation. However, this does not apply to those contacts who do not access TPT services. The private lab and portable computer-assisted radiography X-ray machines were also rolled in; therefore, increasing the access to X-ray could have benefited [27,28]. Previous studies from central India, Ethiopia, Zimbabwe, and South Africa had also observed lack of knowledge as a barrier [20,21,29,30]. Successful contact tracing is critical, which is the first-step care cascade [28]. It requires a healthcare workforce and technologies, both of which were present in the district under PPP. 

However, when it came to treatment initiation, the study revealed an intriguing trend. Despite both government-run centers being suboptimal, they outperformed the PPP district in initiating TB treatment and determining TPT eligibility. There is limited data on TPT-initiation rates among household contacts, while studies among children shows suboptimal rates in India (19–33%) [17,18,19] and varied rates in other countries (32–89%) [21,28,29,31]. Both government-run districts had lower beneficiary load, with one employing a “test and treat” strategy that screened out noninfected contacts. These districts effectively managed drug shortages utilizing local purchasing. These issues undermined healthcare workers’ confidence in initiating TPT [21]. Evidence suggests TPT significantly reduces TB disease among positive-tuberculin skin-test contacts, prompting WHO to advocate ‘test and treat’, but its usefulness in a high-TB-endemic area like Delhi is debated [14,27,32]. Further, low initiation may also be due to patients’ and private providers’ attitude. Beneficiaries’ were concerned about taking TPT and perceive risks when not sick, which was evident among the private providers reluctance to start TPT, aligning with prior reports [20]. Addressing these concerns requires effective counseling and targeted education campaigns [27]. Additionally, generating accurate and reliable data through research can dispel misconceptions and promote understanding of TPT’s benefits. 

Concerns regarding the current TPT drugs (6H regimen) including their long duration, daily intake, compliance, and side effects were raised. The new 3HP regimen may address the majority of them while at the same time being cost-effective and improving completion rates [21,22,27,33]. With regard to TPT-completion information, we opted not to mention its rate. The data extracted from NIKSHAY was deficient mostly, and if ever present, appears highly under-reported. NGO collected data on TPT-completion rates finding, while more robust, that they underscored the poor quality of corresponding NIKSHAY data [7,26].

Private provider engagement in Delhi’s TPT program was limited. While the private provider had distrust in TPT and there was reluctance to initiate treatment. However, the district’s successful partnership with an NGO demonstrated PPP’s potential to enhance TPT efficiency, particularly in initial stages. Leveraging the public–private mix (PPM) and Patient Provider Support Agency (PPSA) models, as seen in other states, could further increase private sector engagement, especially for “Test and Treat” expansion [26]. Engaging the private sector not only in TPT-service delivery but also in planning, communication, and research is vital [34]. Addressing private practitioners’ concerns about Isoniazid resistance and overcoming reluctance to advocate TPT to their patients is a significant step [34].

Our study identified few enablers and barriers to TPT implementation from the provider perspective. Patients and their families had good TB and TPT awareness—fostered by India’s successful TB program—which supported contact tracing and TPT uptake. Dedicated staff, collaborations with NGOs, periodic counselling, and a shorter regimen were key enablers. However, irregular drug supply, procurement challenges, and low national program TPT prioritization remained a major barrier. The drug availability challenge is universal, affecting India and other countries [20,21,22]. Scaling up TPT from IPT necessitates increased financing due to substantial expansion costs [27]. Additionally, despite program inclusion, lower prioritization of TPT, and shortage of human resources burden TPT managing in government centers [13,34]. Dedicated staff recruitment is crucial for effective contact tracing, screening, improving adherence, follow-up, and program execution [21]. Community concerns towards taking TPT when not sick, due to its duration and potential side effects can be addressed through comprehensive education, counselling, regular follow-up, and the involvement of the local community and TB champions [35]. Shorter regimens like 3HP hold promise for enhanced uptake.

Our study offers valuable programmatic implications and potential solutions for optimizing TPT implementation. First, the issue of drug shortages, can be solved through increased financing and central procurement of drugs, enhancing supply chain stability. However, to mitigate stockouts, these should also be a provision of a dedicated budget for local purchase. Second, recruiting and maintaining dedicated TPT staff can address the human resources shortages. Regular training, feedback, supportive supervision, home visits, and community health education facilitated by TB champions should be integral components of this solution. Third, efficient contact tracing and screening are crucial to enroll all household contacts for testing and TPT initiation. While establishing a dedicated TPT workforce is crucial, exploring public–private partnerships (PPP) can provide support in government-run centers until such a workforce is fully operational. Enhancing compliance can be achieved by offering shorter-duration drugs (3HP). Finally, further research on TPT efficacy is needed to reduce skepticism and reluctance among private provider and the community.

## 5. Conclusions

Our study assesses TPT implementation in selected district TB centers in Delhi during the initial phase. While the TPT program offered promises a well-established organizational structures and increased patient awareness, several challenges hindered optimal program execution. Drug shortages, limited budget, human resource constraints, and private provider reluctance emerged as major barriers. However, enablers like dedicated staff, NGO partnerships, and shorter regimens like 3HP proved beneficial. Strategies for improving awareness, engagement, and education are vital for community participation and acceptance. Strengthening human resources through staff recruitment, specialized training, dedicated staffing, and supportive supervision can enhance program efficiency. Collaborating with the private sector, enhanced financing and optimizing drug supply and logistics are crucial steps toward achieving better coverage. All these would contribute to improved health outcomes for individuals and communities affected by tuberculosis, ultimately resulting in the reduction of TB burden. By tackling these challenges and capitalizing on the identified enablers, Delhi can effectively improve its TPT program. This would not only benefit its own population but also contribute significantly to the global fight against this devastating disease.

## Figures and Tables

**Figure 1 tropicalmed-09-00024-f001:**
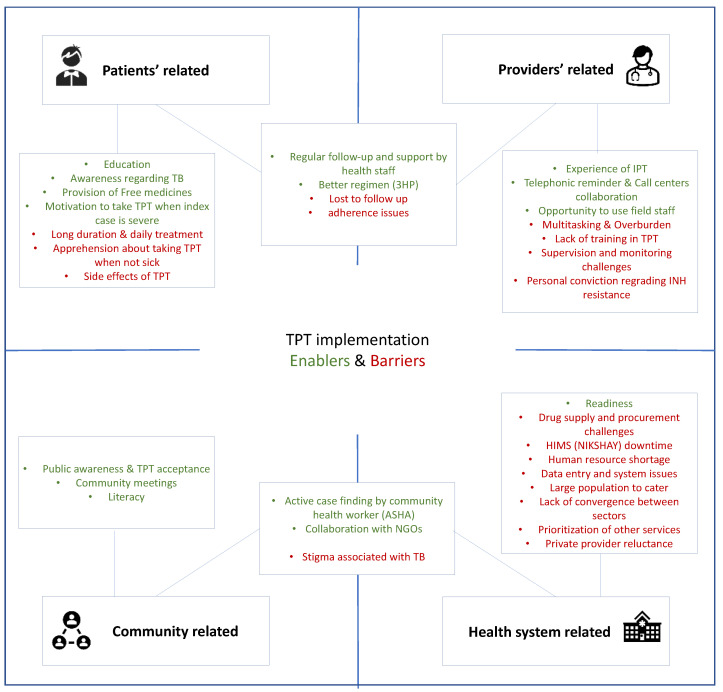
The enablers and barriers for the implementation of TPT in Delhi as perceived by healthcare providers. TB: Tuberculosis, 3HP: weekly regimen of isoniazid-rifapentine for 12 weeks, IPT: Isoniazid preventive therapy, INH: Isoniazid, ASHA: Accredited Social Health Activist, NGO: Non-government organization, HIMS: hospital management information system, NIKSHAY: Indian HIMS for TB patients.

**Table 1 tropicalmed-09-00024-t001:** Cascade of TPT care at three district TB centers in Delhi between January and June 2022.

		**District 1**		**District 2**	**District 3**	**Comments**
	**TPT Implementation**	**Project Mode**		**Govt** ‘**Treat Only**’	**Govt ‘Test and Treat’**	
	**Data source**	NIKSHAY	Project data	NIKSHAY	NIKSHAY	− District 1 had two data source, one by government and one by NGO
**Contact Tracing**	Index patients	1860	2150 ^@^	1062	933	− District with NGO support performed better in contact tracing. − DTO of District 2 and 3 consider untrained staff in initial phase causing incomplete data entry for the deficiencies in the contact tracing, screening and evaluation of TB
	Initial home visit/assessment completed	1811/1860 (97.4%)	1563/2150 (72.7%)	507/1062 (47.7%)	640/933 (68.6%)	
	Index with at least 1 HHC enlisted	1738 (96.0%)	1249 (79.9%)	488 (45.9%)	578 (62.0%)	
	Total HHC enlisted (Avg. HH size:)	7515 (5.1)	4162 (3.7)	1712 (2.6)	1922 (3.1)	
**Screening**	4s Screening completed	6373/7515 (84.8%)	4042/4162 (97.1%)	1582/1712 (92.4%)	1817/1922 (94.5%)	− Symptom screening was sub optimal but district with NGO support reported more than 5% of HHC, which is as per estimations.
	HHC with TB Symptom	346/6373 (5.4%)	55/4042 (1.4%)	32/1582 (2.0%)	51/1817 (2.8%)	
**Evaluation for TB**	Evaluated for TB	166/346 (48.0%)	2323 ^$^	9/32 (28.1%)	18/51 (35.3%)	− Evaluation for TB was poor among all district, we believe it was also due to incomplete data entry. − Government ran districts performed better in initiation of TB treatment
	HHC Diagnosed TB out of those who were evaluated	100/166 (60.2%)	Not recorded	6/9 (66.7%)	10/18 (55.6)	
	Initiated on TB treatment	64/100 (64.0%)	Not recorded	6/6 (100%)	8/10 (80.0%)	
**TPT initiation**	Eligible for TPT *	1163/6273 ^ (18.5%)	3863/4042 ^ (95.6%)	1094/1576 ^ (69.4%)	1189/1807 (65.8%)	− The district following ‘Test and treat’ strategy was expected to have lower eligible HHC, and performed as per expectations. − Eligibility screening was poor in other two districts. − Initiation of TPT was overall poor due to shortage of drugs. Those district which provision of local purchasing performed better − DTO of District 1 consider confusion in initial phase leading to wrong/non data entry for the deficiencies in TPT initiation.
	Provided TPT among those who were eligible *	1054/1163 (90.6%)	2274/3863 (58.9%)	1081/1094 (98.8%)	1162/1189 (97.8%)	
	Provided TPT among those who should eligible ^	1054/6273 (16.8%)	-	1081/1576 (68.6%)	-	
**TPT completion**	Completed TPT	Cannot be assessed	2114/2274 (93.0%)	Cannot be assessed	Cannot be assessed	− TPT-completion rates cannot be assessed as TPT registers were not updated

District 1: Project mode, with government implementing TPT using ‘Treat only’ strategy with support from dedicated staff provided by NGO. District 2: Government implementing TPT using ‘Treat only’ strategy. District 3: Government implementing TPT using ‘Test and Treat’. ^@^ Included Public and Private. ^$^ X-ray of the HHC was completed irrespective of their symptoms. * According to the data source, ^ According to estimates, i.e., those who were screened and not diagnosed TB in district following ‘treat only’ strategy. TB: tuberculosis, TPT: Tuberculosis preventive treatment, NGO: non-government organization, NIKSHAY: Indian hospital management information system for TB patients, DTO: district TB officer, HHC: household contact of Index TB patient, Govt: government.

**Table 2 tropicalmed-09-00024-t002:** Implementation of TPT at selected district TB centers from Delhi, India, using proposed framework.

Domain	Activity	Status *	Comments/Notes
**Organization Structure**	State TPT committee	Established	Need to establish more regular meetings
	TPT implementation plan in annual PIP	Integrated	
	Budget for TPT allocated	Allocated	No separate budget
	Implemented early at all the center	Completed	PPM be utilized at low-performing/high-load center
	Mapping activity, beneficiary, training, drugs and logistics	Completed	Regular mapping needed
	Advocacy plan	Prepared	More active and coverage required
	Mechanism for TBI screening established	Partially established	‘Treat only’ strategy is present in two district while ‘Test and Treat’ in third. ‘Test and Treat’ strategy can be utilized
	Active case finding integrating	Completed	Active case finding supplements
	TPT implemented at private settings	Not completed	
**Human Resource (HR) and Training**	DOT provider/treatment supporter engagement in TPT	Completed	DOT provider is focal person
	HR mapping	Partially Completed	In Project mode district, NGO is supplementing DOT provider. In public mode districts, additional staffing required. May create new care dedicated to TPT
	Induction training	Completed	All the concern staff are trained
	Periodic training	Irregular	Frequent training required, special training for data manager
	Linkage in HMS (Nikshay portal)	Not linked	Contact tracing and TPT register data on NIKSHAY do not comprehend each other Not updated regular basis Improve data updating and alignment
	Mechanism for Review data established	Not linked to HMS	Data is reviewed manually, not embedded in NIKSHAY Deficiency in monitoring, data management, and supportive supervision. Training and support needed
**Community Engagement**	Community volunteers identified and trained	Completed	TB survivors and champions trained
	Advocacy in community	Partially completed	Limited to mass media and IEC activity Sandwiched with other IEC Increase coverage and prioritization required
**Diagnostics**	TBI tests	Partially available	TST available in one district, IGRA not utilized in any center. No provision for outsourcing; explore possibilities
	X-ray	Utilized	In Project mode district, free Xray of all adult cases at close to people home, while in government-run district, only at government health center
**Drugs and Logistics**	Drugs supply and mapping	Initial phase	Improve supply chain Local purchase not sustainable Separate individual beneficiary box with drug for entire duration
	Space for drug storage	Identified and upgraded	
	6H and 3HP drugs	Added in program	Availability is not regular
	TPT adherence mechanisms	Partially established	Adherence by telephonic or pill counts, not home visit, expand mechanisms
	ADR management	Established	
	Logistics for recording and reporting	Adequate	Not in use to expected level
**Private Provider Engagement**	Private provider engagement	Partial	Limited to one center, expand coverage
	Supports for various activities ^	Partially completed	No provision for drug dispensing, TBI screening (IGRA, C-Tb) Expand services

* Assessed as Complete/Partial/Not started. Complete, all centers have positive response; Partial, one center has implemented. ^ Support for contact tracing, TBI screening (IGRA, C-tb), TB diagnosis (X-ray), drug dispensing, counselling, referral, adherence management, advocacy, and community empowerment. IGRA: Interferon-Gamma Release Assays, a blood tests diagnosing Mycobacterium tuberculosis infection, TST: Tuberculin skin tests, ADR: adverse drug reaction, IEC: information education communication, TPT: Tuberculosis preventive treatment, PIP: Program Implementation Plan, PPM: public–private mix, DOT: Directly Observed Treatment.

**Table 3 tropicalmed-09-00024-t003:** Barriers and possible solutions for implementation of TPT among household contacts as perceived by HCW from Delhi, India, 2022–23.

Themes	Categories	Verbatim Quotes
**barrier to TPT**	Not disease	“*No matter how much you counsel and make them understand, they say that they are not sick, so why should they be taking medicines for 6 months*?”—Senior Treatment Supervisor (STS), 45 y, Male (M) with >5 years (y) experiences.
	Drug supply/procurement	“*only the drugs are not in supply. We can do local purchasing, but we don’t have enough funding to purchase drugs for all the beneficiaries*.”—District TB officer (DTO), 40+ y, M, with >10 y experiences).
	Data management/training	“…*there was hindrance that the initial cascade training of the staff was not that effective……resulted in big issue of correct data entry. We then retrained these staffs in data entry and we observe improvement but again this is still prevailing*.”—State Level TB officer), 50+ y, M, with >20 y experiences.
	Drug resistance	“*they also think that if they take drug now, and in future if they develop disease, they might have resistance to the drugs. this is some concern especially among educated population*.”—DTO, 40+ y, M, with >15 y experiences.
	Private provider	“*Convincing them to have treatment for TPT is a challenge especially who are well read and in an era of easy access to Google any other information… there is a reluctance among physicians especially private physicians*…”—State-level TB officer, 50+ y, M, with >20 y experiences.
	Long duration of treatment	“…*there is a sense (among household contacts) that they do not had any disease, so why should take medicine that too for 6 months, it is a long term*.”—Project staff, 30 y, F, with 5 y experiences.
	High patient load/overburden	“*we need manpower, because we have only one tbhv (Health Visitor), which is being used for distribution of drugs and ration food packs, for active contact tracing, bank detail DBT (direct bank transfer), and TPT implementation. so work load is high*.” DTO, 40+ y, M, with >10 y experiences).
	Side effects	“*lost to follow up has also been there who have had adverse reactions. Many said they were nervous, itchy, feeling restless, they did not complete (TPT*).”—Project staff, 30 y, F, with 5 y experiences.
**Possible solutions to TPT**	Counselling	“*It mainly depends on our counselling. As we explain it to them that currently, there is only one patient in the family and in future the other members are too vulnerable for the same. We find that patients come to take the medicine and generally get agree to it*.”—State-level TB officer, 50+ y, M, with >20 y experiences.
	Staff recruitment	“*we required new staff*”—DTO, 40+ y, M, with >10 y experiences.
	Streamlining contact tracing	“*As when the patient visits the center, the details of their household contacts are taken. Then it is conveyed to the patient that the health volunteer will have meeting to the contacts*.”—DTO, 50+ y, M, with >20 y experiences).
	Weekly regimen/shorter regimen	“*3HP is good as its course is for (only) three months and patients do not have to suffer or take medication for longer time. Its good also because one has to take medicine only once a week*.”—Project staff, 48 y, F, with >10 y experiences.
	Awareness/stigma reduction	“*(among awareness programs) we should focus on the community level that everyone should be aware about this disease and there should be no stigma related to this*.”—Project staff, 35 y, M with 3 y experiences.
	Local purchasing	“*there was shortage many times but it’s not a costly drug so we do local purchasing and provide to the patients*.”—Medical officer, 45 y+, M with 15 y+ experiences).

**Table 4 tropicalmed-09-00024-t004:** Possible solutions and recommendations by health providers for better implementation of TPT.

Challenge	Solutions
Drugs unavailability, erratic drug supply, procurement challenges	Central procurement of drugs Ensure timely availability of drugs Local purchasing with dedicated budget
Apprehension about initiating TPT Adherence and lost to follow-up Side effects	Educate and raise awareness about TPT Enhanced communication strategies Improved monitoring and home visits Technology-driven communication Utilizing CHW (ASHA) Evidence generation
Limited availability of TBI diagnostics	Capacity building PPSA and PPM
Long duration amd daily treatment INH resistance with monotherapy (6H)	Consider other drugs with shorter duration (3HP)
Overburdened human resources (HR) Data entry and system issues TPT data on HIMS (NIKSHAY) not available/unreliable	Dedicated HR for TPT Private sector involvement Regular supportive supervision Weekly and fortnightly Monitoring
Not in private sector, private provider reluctance utility of TPT	Utilize PPM provisions Evidence generation
Stigma associated with TB	Strengthened community engagement TB champions

6H: six month isoniazid regimen, 3HP: weekly regimen of isoniazid-rifapentine for 12 weeks, HIMS: hospital management information system, NIKSHAY: Indian HIMS for TB patients, CHW: Community Health Worker, ASHA: Accredited Social Health Activist, PPSA: Patient Provider Support Agency, PPM: public–private mix.

## Data Availability

Requests to access these data should be sent to the corresponding author.

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
