# Peer review of "Situation Analysis of Early Implementation of Programmatic Management of Tuberculosis Preventive Treatment among Household Contacts of Pulmonary TB Patients in Delhi, India"

_tropicalmed, 2024, doi:10.3390/tropicalmed9010024_

Round 1

Reviewer 1 Report

Comments and Suggestions for Authors

The study identified differences among the three "Chest Clinics" chosen for the study.  Data entry (incomplete entry) was an issue identified.

Evaluation of these programs should be monitored in a regular fashion to ensure that eligible patients receive preventative therapy for tuberculosis.

It would be of interest to look at the numbers of household contacts that progressed to tuberculosis over the previous 2 years comparing the three "Chest Clinics" to see if there are any differences in outcome.

Author Response

Thank you for your insightful comments on our manuscript. We appreciate your keen observation regarding the differences among the three "Chest Clinics" chosen for our study. Recognizing these distinctions is crucial, and we acknowledge their potential impact on the overall outcomes.

Your point about incomplete data entry is well taken, and we have duly noted this concern in our manuscript.

The suggestion to monitor the evaluation of these programs regularly for the administration of preventative therapy to eligible patients is well- appreciate. We understand the importance of continuous monitoring to enhance the effectiveness of TB prevention efforts.

Regarding the recommendation to explore the numbers of household contacts progressing to tuberculosis over the previous 2 years, we find this suggestion valuable. Unfortunately, this specific data is currently unavailable to us and may not directly contribute to understanding the recently initiated 'TB preventive treatment' services, which is the primary focus of our study. However, we acknowledge the importance of this aspect and will make efforts to incorporate such analyses if the relevant data becomes accessible in the future.

We are grateful for the reviewer's insightful suggestion to provide the supplementary material with the numerical values associated with the TPT cascade. We have readily incorporated the reviewer's recommended addition to Figure S2.

We sincerely appreciate your constructive feedback, which undoubtedly enriches the quality and depth of our research.

Reviewer 2 Report

Comments and Suggestions for Authors

This is an original article that accurately describes the implementation and development of a tuberculosis (TB) contact tracing program and TB preventive treatment cascade (TPT) in Delhi (India). India is a high burden TB country and TPT is still a challenge worldwide, due to the difficulties encountered in health services, related to providers and people investigated/TB contacts. The authors used a difficult methodology (convergent design) but which was appropriate for the work they proposed.

The methods are carefully described, and the Discussion is very relevant to the results.

My suggestion to the authors would be to present in the supplementary material the TPT cascade (Figure S2. Flow chart showing the assessment for TPT eligibility according to guidelines for Programmatic Management of TB Preventive Treatment in India 2021) with the numerical values ​​of the contacts investigated. I believe that one could have a more realistic idea of ​​the performance of the established program.

Author Response

We sincerely appreciate the reviewer's positive feedback on our original article regarding the TB contact tracing program and TPT cascade in Delhi. We are particularly grateful for their recognition of the article's originality, relevance of the subject matter, and careful description of methods and discussion.  

We completely agree with the reviewer's suggestion to include the numerical values of investigated contacts within the TPT cascade figure (Figure S2) in the supplementary material, which has been added in revised version. This would undoubtedly enhance the transparency and clarity of the program's performance.  

We appreciate your meticulous evaluation and valuable suggestions.  

Reviewer 3 Report

Comments and Suggestions for Authors

MDPI Revision

Tropical Medicine and Infections Disease

"Situation analysis of early implementation of programmatic management of Tuberculosis Preventive Treatment among household contacts of Pulmonary TB patients in Delhi, India"

The article is relevant, but it needs to enrich the motivation and analysis of the study supported by technical models. I suggest acceptance of the article through major revisions.

1 – The abstract should provide better detail of the article, perhaps indicating the main results of the analysis performed, including limitations and future research.

2 - The main problem of the work concerns the literature review. Considering the relevance of the topic and the impact of this journal, a literature review should be considered, trying to reflect the importance of the proposed analysis with strategical implementation in health environment. Also, I suggest the inclusion and explanation of the main related works, based in the use of technical models as decision-making aid in complex problems, using mathematical and statistical modeling.

3 – I realized a lack of mathematical basis and quantitative evaluation to support the analysis and data process to support the idea and paper motivation.

4 – Considering a set of stakeholders, which modeling have you been used to achieve the consensus between the decision-makers?

5 – The introduction section needs to provide the paper structure.

6 - The conclusion is too short. Provide more detail about the study's limitations and future proposals for a more accurate analysis.

7 – Provide a general review of English grammar.

Comments on the Quality of English Language

Provide a general review of English grammar.

Author Response

Thank you for your insightful comments on our manuscript. We appreciate your keen observation and constructive feedback. Please find point wise reply to your comments.

1. We fully agree with the reviewer's suggestion to strengthen the abstract. We have revised it to offer a concise yet informative overview of the article's key findings. Although constrained by the 200-word limit, we believe this revised abstract provides a more complete picture of our work.

2. Recognizing the issue of the literature review raised, we have expanded this section incorporate additional relevant studies. We now have added a section on cascade care analysis. We believe this improves the introduction’s comprehensive overview of existing research in the field along with reflection of proposed analysis.

3&4. We acknowledge the reviewers' concerns about the limited quantitative data and modeling in the current manuscript. While we agree that such approaches would be valuable, our study was designed as a rapid assessment using a mixed-methods convergent design to understand the underlying reasons for the challenges in TPT implementation. We believe that adding extensive quantitative data at this stage would have significantly diverted from our study's primary objective of investigating the "why" and "what" behind the observed implementation issues. However, we recognize the potential value of modeling the data and plan to explore this further in a future manuscript. We believe the current results provide sufficient evidence for the study's initial objectives.

5. Thank you for pointing this out. We have revised the introduction section to offer a more explicit overview of the paper's structure. This includes expanded background information, a stronger emphasis on the importance of TPT, a clear explanation of the proposed cascade care analysis, and a concise summary of relevant previous research. We hope this revised structure provides readers with a better understanding of the study's context and objectives.

6. We appreciate the feedback. We have expanded the conclusion section to provide more specific insights. 

7. Two authors have thoroughly reviewed the revised manuscript for grammatical and spelling errors, ensuring its accuracy and clarity.

We believe these revisions have addressed the reviewers' concerns and significantly improved the manuscript. We are grateful for their valuable feedback. 

Reviewer 4 Report

Comments and Suggestions for Authors

Overall, congratulations to the co-authors for conducting this important study on the pragmatics of TPT implementation. The manuscript has much to recommend it - an important, clearly defined topic, appropriate data collection and analysis, and reasonable conclusions. I suggest the writing itself could be stronger / clearer, with suggestions below:

Minor comments:

Spelling / grammar / writing - there are minor issues throughout. E.g., line 240 "One general theme that emerge" should be "One general theme that emerged". These are minor, but please take the opportunity to address. 

Line 120 - What Nikshay is gets explained in line 143, should rather have the explanation at the first use.

Lines 164-165 - Reads odd to have a sub-heading with only "see elsewhere" under the sub-heading.

Lines 214-215 - Unnecessary repetition of information already covered in lines 123-125.

In the results - I suggest that it is more intuitive to the reader if the order of Table 1 and 2 are presented in the opposite order; i.e., that the reader sees the content in table 2 first. That is sort of 'what is going on'. Then the content in table 1 which is a summary of an operational review that might explain some of 'why' this is going on. And finally the qualitative data that might explain the underlying processes of why this is going on.

Lines 315-317 - the exclusion of patient / survivors' perspectives is substantive. The discussion frames the findings as an exhaustive / comprehensive review of 'the facilitators/barriers' to TPT implementation. Instead, more accurately, these are 'some of the facilitators/barriers', limited to those identified from the provider perspective. An example, the providers said that 'the community' were well-informed about TPT and therefore there was a high demand / a facilitator to role-out. But, almost by definition, providers are only seeing people accessing services. Those who are not accessing services might not be informed about TPT. Draw this into the comparison of your findings to the established literature (lines 340-345). I suggest softening the interpretation of the findings in the discussion to accommodate this.

Lines 346-422 - this is all overly repetitive of the findings and interpretations often speculative. I strongly suggest that this content be included in the findings section and that the discussion be shorter, and instead focus on detailing implications for policy, practice, and future research.

Comments on the Quality of English Language

See comments in overall review. 

Author Response

Thank you for your thorough review and constructive comments. We have carefully considered each point, and here is our response:

1. We acknowledge the minor issues in spelling and grammar throughout the manuscript. The revised version has undergone a meticulous review, addressing and rectifying these concerns.

2. We have relocated the explanation of NIKSHAY to its first use in the manuscript, ensuring clarity for the reader.

3. We have revised the presentation of this sub-heading (Lines 164-165). We hope this adjustment enhances the flow and readability.

4. We have eliminated the unnecessary repetition of information (Lines 214-215), streamlining the presentation for improved coherence.

5. Following your suggestion, we have reversed the order of Tables 1 and 2. Thankyou for this.

6. We have restructured the discussion section, incorporating your feedback. The tone of interpretation has been softened, explicitly acknowledging the limitations regarding patient/survivor perspectives. Additionally, we have minimized speculative content and avoided excessive repetition of findings.

We appreciate your meticulous evaluation and valuable suggestions, which have significantly contributed to refining our manuscript.

Round 2

Reviewer 3 Report

Comments and Suggestions for Authors

The authors presented a new version of the article, following all the suggestions approved in the last review.

Therefore, I suggest accepting the work.

Comments on the Quality of English Language

English is fine, but can have a little improvement in some parts.